# Modulation of the Antibiotic Activity by the *Mauritia flexuosa* (Buriti) Fixed Oil against Methicillin-Resistant Staphylococcus Aureus (MRSA) and Other Multidrug-Resistant (MDR) Bacterial Strains

**DOI:** 10.3390/pathogens7040098

**Published:** 2018-12-10

**Authors:** Yara Faustino Pereira, Maria do Socorro Costa, Saulo Relison Tintino, Janaína Esmeraldo Rocha, Fábio Fernandes Galvão Rodrigues, Maria Karine de Sá Barreto Feitosa, Irwin Rose Alencar de Menezes, Henrique Douglas Melo Coutinho, José Galberto Martins da Costa, Erlânio Oliveira de Sousa

**Affiliations:** 1Technology Center, Laboratory Analysis Physical Chemistry of Food, Faculty of Technology Cariri, Juazeiro do Norte 63041-190, Brazil; yarafaust13@gmail.com (Y.F.P.); karine_bf@hotmail.com (M.K.d.S.B.F.); erlaniourca@hotmail.com (E.O.d.S.); 2Department of Biological Chemistry, Laboratory of Microbiology and Molecular Biology, Regional University of Cariri, Crato, CE 63105-000, Brazil; corrinha_live@yahoo.com.br (M.d.S.C.); saulorelison@gmail.com (S.R.T.); janainaesmeraldo@gmail.com (J.E.R.); 3Department of Biological Chemistry, Laboratory of Research in Natural Products, Regional University of Cariri, Crato, CE 63105-000, Brazil; fabiogalvao01@hotmail.com (F.F.G.R.); galberto.martins@gmail.com (J.G.M.d.C.); 4Department of Biological Chemistry, Laboratory of Pharmacology and Molecular Chemistry, Regional University of Cariri, Crato, CE 63105-000, Brazil; irwinalencar@yahoo.com.br

**Keywords:** buriti, fatty acid, antibacterial, aminoglycosides

## Abstract

*Mauritia flexuosa* (buriti) is a typical Brazilian palm tree found in swampy regions with many plant forms. The fruit has various purposes with the pulps to the seeds being used for ice creams, sweets, creams, jellies, liqueurs, and vitamin production. A physicochemical characterization of the fixed pulp oil and its antibacterial and aminoglycoside antibiotic modifying activity against Gram-positive and Gram-negative multiresistant bacterial strains were performed using broth microdilution assays. Physical properties, such as moisture, pH, acidity, peroxide index, relative density, and refractive index, indicated oil stability and chemical quality. In the GC/MS chemical composition analysis, a high content of unsaturated fatty acids (89.81%) in relation to saturated fatty acids (10.19%) was observed. Oleic acid (89.81%) was the main fatty acid identified. In the antibacterial test, the fixed oil obtained the Minimum Inhibitory Concentration (MIC) ≥ 1024 μg/mL for all standard and multiresistant bacterial strains. The synergic effect of fixed pulp oil combined was observed only in *Staphylococcus aureus* SA–10, with an MIC reduction of the gentamicin and amikacin by 40.00% and 60.55%, respectively. The data indicates the *M. flexuosa* fixed oil as a valuable source of oleic acid and modulator of aminoglycoside activity.

## 1. Introduction

The increase in microorganisms resistant to clinically important antimicrobials, such as aminoglycosides, have been challenging science and causing serious public health risks [1]. Severe toxicity is one of the aminoglycoside resistance problems associated with high doses or chronic treatment leading to ototoxicity and/or nephrotoxicity [2].

Aminoglycosides are antibiotics with a mechanism of action based on protein synthesis, where by binding to the 30S prokaryotic ribosome, these prevent adequate mRNA translation [2]. Despite a wide bactericidal spectrum and benefits in the treatment of many infectious disorders that exist, bacterial resistance to aminoglycosides has become a problem in recent decades [3].

Due to aminoglycoside resistance, many plants have been studied not just for their antibacterial activity, but also for their modifying action. Natural plant products can alter antibiotic effects by improving or decreasing their activity such that these products can be a viable alternative for the resistance issue [4,5].

Different natural products, including fixed oil metabolites, have potentiated the antibiotic effect against certain bacteria [2,4,5,6]. This strategy, known as herbal shotgun or synergistic multi-effect targeting, refers to the use of plants alongside drugs in an approach using mono/multi extracts or combined oils affecting several microorganismal targets at the same time with therapeutic components collaborating synergistically in an agonistic manner [3].

*Mauritia flexuosa* is one of the most interesting palm trees in Brazil, common in the Pará, Amazonas, Amapá, Rondônia, Goiás, Distrito Federal, Bahia, Minas Gerais, Mato Grosso, Ceará, and Maranhão states. The fruit has diverse purposes, with the pulp being used in ice creams, jams, creams, jellies, liqueurs, and vitamins [7]. This palm tree is also used for edible oil extraction and its nutritional worth varies with seasonality and the extraction process [8].

The extracted oil has an orange coloration and is normally extracted in an artisanal manner [7,8]. However, mechanical extraction is the most common method used in cooperatives and industries, this being less aggressive than the artisanal process, which probably minimizes oil oxidation and enables a longer shelf life [9]. Due to its chemical composition, the oil is of interest because of its pharmacological potential [10].

In this sense, the objective of this study was to evaluate the physicochemical characterization of the *M. flexuosa* fixed pulp oil obtained via cold pressing, as well as its antibacterial activity in isolation and in association with antibiotics.

## 2. Materials and Methods

### 2.1. Plant Material and Botanical Identification

Fruits of *Mauritia flexuosa* (buriti) were collected (Sítio Lameiro) in an area of the Chapada do Araripe, Municipality of Crato, Ceará, Brazil. An exsiccate (#9710) of the species is found in the Herbarium Caririense Dárdano Andrade Lima (HCDAL) of the Regional University of Cariri (URCA).

### 2.2. Pulp and Fixed Oil Acquisition

The pulp was obtained by manually removing the internal mesocarp from the fruit, and then crushed in a blender and lyophilized. The fixed oil was obtained using the mechanical extraction method with discontinuous hydraulic pressing using 500 g of *M. flexuosa* almonds. The sample was added to a stainless-steel cylinder and pressed for approximately 2 h, the pressure of which was recorded by a manometer at 15 T. The collected fixed oil was obtained yielding 37.20% of the crude material and stored in a hermetically sealed amber flask and kept refrigerated.

### 2.3. Physicochemical Characterization

Physicochemical analysis was performed on the fixed oil with regard to the following parameters: water content, pH, acidity (as oleic acid), relative density, peroxide index, and refractive index at 40 °C [9].

### 2.4. Fatty Acid Analysis

Fatty acids were determined indirectly using their corresponding methyl esters. The oil (0.2 g) was saponified for 30 min under reflux with a potassium hydroxide solution in methanol, following the method described by Hartman and Lago [11]. After adequate treatment and pH adjustment, the free fatty acids were methylated with methanol via acid catalysis in order to yield the respective methyl esters.

The analysis of volatile constituents was carried out in a GC/MS HP model 5971 using the non-polar fused silica column DB-1 (30 m × 0.25 mm i.d., 0.25 μm film), eluted with helium gas at 8 mL/min and with split mode. Injector and detector temperatures were set to 250 °C and 200 °C, respectively. The column temperature was programmed to rise from 35 °C to 180 °C at 4 °C/min, and then from 180 °C to 250 °C at 10 °C/min. Mass spectra were recorded from 30 to 450 *m/z*, with an electron beam energy of 70 eV. The injected sample was 1 μL of 5 mg/mL of the oil solution fixed in acetone.

The individual components were identified by matching their mass spectra with those of the database using the library constructed using the spectrometer Wiley 229 and NIST 08 using retention indices (IR) as a pre-selection [12], as well as by visually comparing standard fragmentation to that reported in the literature [13].

### 2.5. Antibacterial Analysis

#### 2.5.1. Strains Utilized

Standard and multiresistant strains of bacteria were used in the analyses. The standard strains were: *Staphylococcus aureus* SA-ATCC 6538, *Bacillus cereus* BC-ATCC 33018, *Escherichia coli* EC-ATCC 10536, *Pseudomonas aeruginosa* PA-ATCC 9027, *Klebsiella pneumoniae* KP-ATCC 10031, *Shigella flexneri* SF-ATCC 12022, and *Proteus vulgaris* PV-ATCC 13315 and all were obtained from the American Type Culture Collection (ATCC). The multiresistant strains were: *S. aureus* SA-10 and *E*. *coli* EC-06 with source and resistance profiles identified in Table 1. These were maintained in a blood agar base (Laboratory Difco Ltd.,São Paulo, Brazil) and cultured at 37 °C for 24 h in Heart Infusion Agar (HIA, Difco. Laboratories Ltd., São Paulo, Brazil).

#### 2.5.2. Antibiotics

Drugs used in the tests were the aminoglycosides amikacin and gentamicin (Sigma Co., St. Louis, MO, USA). All drugs were diluted in sterile water to a concentration of 5000 μg/mL.

#### 2.5.3. Minimum Inhibitory Concentration Test

For the minimum inhibitory concentration (MIC) assays, eppendorfs were prepared with 100 μL of the inoculum and 900 μL of the Brain Heart Infusion (BHI) culture medium in a concentration of 10% in 96-well plates that were filled in the numerical sense by adding 100 μL of this solution into each well. Subsequently, serial microdilution with 100 μL of the tested substance was performed, varying in concentrations from 1024 to 1.0 μg/mL. The plates were taken to the incubator for 24 h at 37 °C. To read the bacterial MIC, 20 μL of resazurin was added to each well, and after 1 h, the color change of the wells was observed, where the modification of the blue to red coloration corresponds to microbial growth and the permanence in blue corresponds to the absence of growth, as established by The Clinical & Laboratory Standards Institute (CLSI) [14].

#### 2.5.4. Antibiotic Activity Modifying Effect

The evaluation of the fixed oils as an antibiotic activity modifier was performed according to Coutinho et al. [4]. The tests were performed in triplicate. For each eppendorf, 1162 μL of 10% BHI was used, with 150 μL of the inoculum of each strain and the tested substance with a volume corresponding to a sub-inhibitory concentration (MIC/8 = 128 μg/mL). Controls were prepared with only 1350 μL of BHI (10%) and 150 μL of bacterial suspension. The plates were filled in numerical order and each well received 100 μL of solution. The microdilution was performed with 100 μL of each antibiotic up to the penultimate well and the final volumes were discarded. The plates were then incubated at 37 °C for 24 h and read through the addition of resazurin.

### 2.6. Statistical Analysis

The data analysis was performed using the statistical program GraphPad Prism 5.0. The data were analyzed through a two-way ANOVA test, using as the central data the geometric mean of the triplicates and the standard deviation of the mean. Subsequently, a Bonferroni post hoc test was performed (where *p* < 0.05 was considered significant).

## 3. Results and Discussion

### 3.1. Oil and Fatty Acid Physicochemical Profile

The physicochemical characterization results are shown in Table 2. The moisture content obtained a value of 0.30%, which represents the minimum percentage of uncombined water (˃1%), suggesting a quality oil with greater durability [15].

The 1.76% acid value reflects stability against neutralization and the peroxide index value, 4.00 meq/kg, indicates greater resistance to oxidation. The pH obtained a value of 4.54, with this parameter being important as it indicates an inhibitory range of several bacterial strains [15]. The refractive index corresponded to 1.46, where this criterion is widely used for oil quality and identity analysis. The density found via direct measurement was 0.304 (g/cm³).

GC/MS spectral analysis allowed for the identification of the *M. flexuos*a fixed oil composition percentage of fatty acids (Table 3 and Figure 1). The fixed oil was characterized by a higher content of unsaturated fatty acids (89.81%) than saturated fatty acids (10.19%). In the oil, oleic acid (89.81%), a monounsaturated fatty acid, was identified as the main fatty acid.

Corroborating with these results, Darnet et al. [16] found oleic (75.7%) and palmitic (18.9%) acids at greater quantities in the oil’s chemical composition. Similarly, Silva et al. [8] found oleic (74.0%) and palmitic (16.7%) acids as the main components of this oil. Melo et al. [17] also observed unsaturated fatty acids as the major component.

### 3.2. Antibacterial and Antibiotic Modifying Activity

In the antibacterial activity test, the results showed high MIC values ≥1024 μg/mL for all standard and multiresistant bacterial strains (Table 4). High values of MIC (512 μg/mL) were also observed for the fixed oil of the pulp extracted with hexane against the several strains of bacteria analyzed [18]. The results obtained may be related to the absence of a higher concentration of metabolites, such as unsaturated fatty acids, which have been shown to have antimicrobial activity [2,19].

On the other hand, phenolic extracts of the leaves, trunk, and fruit of *M. flexuosa* demonstrated antimicrobial activity against some pathogenic bacteria with low MIC values and the best results were found for leaf extracts. The possible cause was attributed to the phenolic composition between the parts of the species [19].

Fixed oil activity against many Gram-positive and Gram-negative strains have been previously demonstrated. The fixed “pequi” pulp oil demonstrated growth inhibition of standard and multiresistant bacterial strains [2,6]. The apple seed oil demonstrated activity against *E. coli* and *S. aureus* standard strains [20].

Table 5 shows the MICs of the antibiotics and the synergic effects of the fixed oil in association with aminoglycoside antibiotics. The MICs of the antibiotics to bacteria strains were in the range of 256 to 20 μg/mL and their MIC decreased in presence of the fixed oil. The fixed oil showed weak antibacterial activity but presented a synergetic effect for antibiotics in association. The synergistic effect expressed was the potentiation of gentamicina and amikacin in *S. aureus* SA-10 with MIC reductions of 40.00% and 60.55%, respectively. These results were observed for the two antibiotics using subinhibitory concentrations, results which were absent against *E. coli* EC-06. Therefore, the oil’s modifying activity over aminoglycoside antibiotics varied according to the antibiotic type associated with the fixed oil and the bacterial strain analyzed.

The synergistic effect of fixed oils from both animals and plants in association with aminoglycoside antibiotics, similar to that observed in the present study, has been shown previously. The fixed “pequi” oil, for example, potentiated the activity of gentamicin, kanamycin, amikacin, and neomycin in a range of 87.5 to 99.8% against *S. aureus* standard and multiresistant strains [2]. The *Rhinella jimi*, an amphibian, fixed skin oil significantly enhanced the antibiotic activity against multiresistant *E. coli* when associated with amikacin [5].

It is believed that both the antibacterial potential and the antibiotic modifying activity attributed to fixed oils is, at least in part, associated with the fatty acids present in their composition since some fatty acids have already been shown to be able to improve antibiotic activity and to inhibit bacterial growth [2,21].

It is reported that the potential of fixed oils to act as antibacterial or antibiotic modifiers is in part associated with the detergent property of fatty acids against the amphipathic structure of bacterial cell membranes [2]. In this sense, the synergistic effect observed against *S. aureus* may be associated with detergent properties of fatty acids. The literature reports that long-chain unsaturated fatty acids, such as oleic and palmitic, demonstrate antibacterial activity and the conjugated use of fatty acids and peptides or antibiotics have potentiated antibiotic activity due to increased membrane permeability [22,23].

This ability to solubilize membrane components (lipids and proteins) creates gaps in this structure that will affect metabolic processes essential for bacterial cell energy acquisition, such as the electron transport chain and oxidative phosphorylation. Membrane damage can also lead to nutrient absorption difficulties, enzymatic activity inhibition, and toxic peroxidation [24]. Additionally, the presence of hydrophobic compounds in fixed oils may increase the cell’s permeability to antibiotics resulting in higher efficiency and reducing the minimum concentration required for the antibiotic to act against the bacterium [1,2].

There is also the possibility that fatty acids act on bacterial resistance mechanisms, where linoleic and oleic acids have been observed to interfere with MsrA pump activity by reducing ethidium bromide efflux [25]. 

## 4. Conclusions

The data obtained shows the *M. flexuosa* fixed pulp oil to be a valuable source of oleic acid and indicates an aminoglycoside antibiotic modulatory potential. Therefore, it is suggested that the pulp be used as a source for fixed oil acquisition to study its antibacterial properties in isolation as well as in association with antibiotics.

## Figures and Tables

**Figure 1 pathogens-07-00098-f001:**
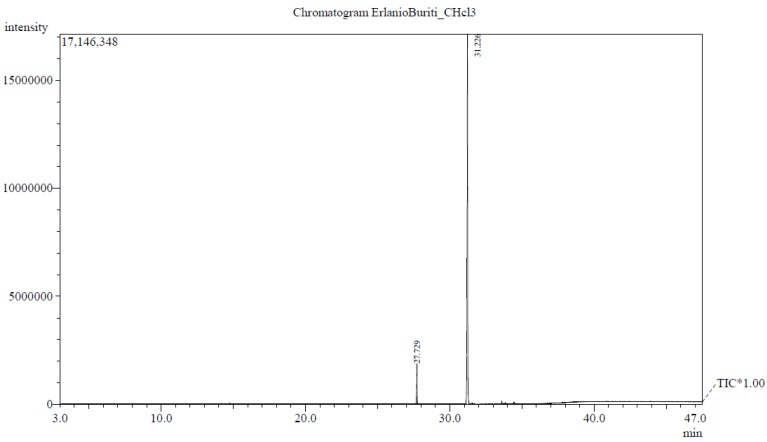
Chromatogram of the GC-MS analysis of the fixed oil of the pulp of *Mauritia flexuosa.* (more information in Appendix A).

**Table 1 pathogens-07-00098-t001:** Bacterial source and antibiotic resistance profile.

Bacteria	Source	Resistance Profile
*S. aureus* (SA-10)	Surgical wound	Cephalothin, Cephalexin, Cefadroxil, Oxacillin, Penicillin, Ampicillin, Ampicillin + Sulbactam, Amoxicillin, Moxifloxacin, Ciprofloxacin, Levofloxacin, Erythromycin, Clarithromycin Azithromycin, and Clindamycin
*E. coli* (EC-06)	Surgical wound	Cephalothin, Cephalexin, Cefadroxil, Ceftriaxone, Cefepime, and Ampicillin + Sulbactam

**Table 2 pathogens-07-00098-t002:** Physico-chemical properties of the fixed oils of the pulp of *M. flexuosa*.

Physico-Chemical Properties	Values
Water contente (% p/p)	0.30 ± 0.50
pH	4.54 ± 0.90
Acidity (as oleic acid %)	1.76 ± 0.85
Relative density (g/cm³)	0.304 ± 0.05
Peroxide índex (meq/Kg)	4.00 ± 1.00
Refractive index (40 °C)	1.46 ± 0.50

Results are expressed with means ± S.E.M. (*n* = 3) of experiments performed in triplicate.

**Table 3 pathogens-07-00098-t003:** Fatty acids identified in the fixed oil of the pulp of *Mauritia flexuosa*.

Order	Constituents	*RI (Min)	%
	*Saturated*		10.19
1	Palmitic acid (C16:0)	27.72	10.19
	*Unsaturated*		*89.81*
2	Oleic acid (C18:1)	31.22	89.81
Total identified			100.00

* Relative retention indices (Adams, 2007).

**Table 4 pathogens-07-00098-t004:** Minimum inhibitory concentration values (MIC, μg/mL) of the fixed oil of the pulp of *Mauritia flexuosa*.

Bacterial Strains	MIC (µg/mL)
*Proteus vulgaris* PV–ATCC 13315	≥1024
*Klebsiella pneumoniae* KP–ATCC 10031	≥1024
*Shigella flexneri* SF–ATCC 12022	≥1024
*Pseudomonas aeruginosa* PA–ATCC 9027	≥1024
*Escherichia coli* EC–ATCC 10536	≥1024
*Escherichia coli* EC–06	≥1024
*Bacillus cereus* BC–ATCC 33018	≥1024
*Staphyloccus aureus* SA–ATCC 6538	≥1024
*Staphyloccus aureus* SA–10	≥1024

**Table 5 pathogens-07-00098-t005:** MIC values (µg/mL) of aminoglycosides with and without the fixed oil of the pulp of *Mauritia flexuosa*.

Antibiotics	*Staphylococcus aureus* SA-10			*Escherichia coli* EC-06		
	MIC Alone	MIC Combined	Reduction MIC %	MIC Alone	MIC Combined	Reduction MIC %
Amikacin	256 ± 0.00	101 ± 4.95	60.55	64 ± 0.00	64 ± 0.00	0
Gentamicin	20 ± 2.22	12 ± 1.49	40.00	32 ± 0.00	32 ± 0.00	0

Values represent the geometric mean ± MSE (mean standard error).

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
