# Peer review of "Modulation of the Antibiotic Activity by the Mauritia flexuosa (Buriti) Fixed Oil against Methicillin-Resistant Staphylococcus Aureus (MRSA) and Other Multidrug-Resistant (MDR) Bacterial Strains"

_pathogens, 2018, doi:10.3390/pathogens7040098_

Round 1

Reviewer 1 Report

·         Why M. flexuosa pulp is only used in this study, though there are some reports that the peels extract of this plant has a better antibacterial activity than pulp extract. See the reference:  de OLIVEIRA AI, CABRAL JB, MAHMOUD TS, do NASCIMENTO GN, da SILVA JF, PIMENTA RS, de MORAIS PB. In vitro antimicrobial activity and fatty acid composition through gas chromatography-mass spectrometry (GC-MS) of ethanol extracts of Mauritia flexuosa (Buriti) fruits. Journal of Medicinal Plants Research. 2017 Oct 25;11(40):635-41.

·         The present study is very similar to research articles entitiled “Chemical composition and antibacterial activity of fixed oils of Mauritia flexuosa and Orbignya speciosa associated with aminoglycosides. published in European Journal of Integrative Medicine. 2018 Oct 1;23:84-9. By Nobre CB, de Sousa EO, de Lima Silva JM, Coutinho HD, da Costa JG group.

·         What is the novelty of this study? There is large number of research similar to this?

1.      Nobre CB, de Sousa EO, de Lima Silva JM, Coutinho HD, da Costa JG. Chemical composition and antibacterial activity of fixed oils of Mauritia flexuosa and Orbignya speciosa associated with aminoglycosides. European Journal of Integrative Medicine. 2018 Oct 1;23:84-9.

2.      Koolen HH, da Silva FM, Gozzo FC, de Souza AQ, de Souza AD. Antioxidant, antimicrobial activities and characterization of phenolic compounds from buriti (Mauritia flexuosa L. f.) by UPLC–ESI-MS/MS. Food Research International. 2013 May 1;51(2):467-73.

3.    Batista JS, Olinda RG, Medeiros VB, Rodrigues CM, Oliveira AF, Paiva ES, Freitas CI, Medeiros AD. Antibacterial and healing activities of buriti oil Mauritia flexuosa L. Ciência Rural. 2012;42(1):136-41.

4.      de OLIVEIRA AI, CABRAL JB, MAHMOUD TS, do NASCIMENTO GN, da SILVA JF, PIMENTA RS, de MORAIS PB. In vitro antimicrobial activity and fatty acid composition through gas chromatography-mass spectrometry (GC-MS) of ethanol extracts of Mauritia flexuosa (Buriti) fruits. Journal of Medicinal Plants Research. 2017 Oct 25;11(40):635-41.

·         If authors performed comparative study of antibacterial activity of different parts of the plants, the effectiveness of the study will surely be enhanced.

·         Table 3: GC/MS spectral analysis was performed to identify the phytochemicals constituents of the M. flexuosa fixed oil using the library NIST 08. Authors have just shown the table, but not the GCMS spectra. Authors must provide the GC/MS spectra that support the table 3. What are the other components present in pulp extract?

·         Why authors have chosen Aminoglycoside antibiotics? Though Aminoglycoside antibiotics display bactericidal activity against Gram-negative aerobes and some anaerobic bacilli where resistance has not yet arisen but generally not against Gram-positive.  

·         The synergistic effects of oil with antibiotics are not clear. How many folds the efficacy of antibiotics increased in association with oils.  

·         Authors did not mention the mechanism and mode of action of M. flexuosa pulp oils.

·         Scanning electron or transmission electron microscopic analysis can described the mechanism of pulp oils.

·         Authors only used MIC to assess the antibacterial test, though for plants extract, diffusion methods showing clear zone of inhibition is more acceptable.

Author Response

REVIEWER 1

Comments and Suggestions for Authors

·  Why M. flexuosa pulp is only used in this study, though there are some reports that the peels extract of this plant has a better antibacterial activity than pulp extract. See the reference:  de OLIVEIRA AI, CABRAL JB, MAHMOUD TS, do NASCIMENTO GN, da SILVA JF, PIMENTA RS, de MORAIS PB. In vitro antimicrobial activity and fatty acid composition through gas chromatography-mass spectrometry (GC-MS) of ethanol extracts of Mauritia flexuosa (Buriti) fruits. Journal of Medicinal Plants Research. 2017 Oct 25;11(40):635-41.

Despite the possibility of the antibacterial activity of other parts of the buriti, fact that some studies already inform, the objective of the present work was to centralize in the chemical characteristic and the antibacterial activity of the fixed oil of the pulp due to the relevance that several communities have given the extraction and commercialization the same. However, there was no study on the chemical quality and antibacterial activity of this oil, which aroused the interest of reproducing the extraction mechanism of the oil and performed analyzes. The results come together with the other works to expand the studies on this species.

·  The present study is very similar to research articles entitiled “Chemical composition and antibacterial activity of fixed oils of Mauritia flexuosa and Orbignya speciosa associated with aminoglycosides. published in European Journal of Integrative Medicine. 2018 Oct 1;23:84-9. By Nobre CB, de Sousa EO, de Lima Silva JM, Coutinho HD, da Costa JG group.

We agree that most of the proposed activities actually have similarities, but in the work of Nobre et al. 2018 was emphasized the fixed oil of buriti extracted with organic solvent that is impossible to be commercialized and used for food purposes etc., while in this work the proposal is to analyze the oil extracted without using solvent. Differently, in this work a physical-chemical study of the fixed oil was presented to certify the chemical quality.

 ·   What is the novelty of this study? There is large number of research similar to this?

It is our knowledge about the existence of several works with this species for various purposes, exploring various other parts and even the fixed oil. However, the present work has a differential proposal when proposing a physical-chemical study to the antibacterial of the fixed oil obtained by hydraulic pressing unlike the others that use oil extracted with organic solvents (ether or hexane). Organic solvent extraction may promote oxidative damage to the oil and may not be used for marketing purposes.

Our research reached its purpose when demonstrated by the physical-chemical analysis that the oil has a characteristic that guarantees durability and chemical stability and verified its antibacterial and modifying capacity of antibiotic activity.

1.      Nobre CB, de Sousa EO, de Lima Silva JM, Coutinho HD, da Costa JG. Chemical composition and antibacterial activity of fixed oils of Mauritia flexuosa and Orbignya speciosa associated with aminoglycosides. European Journal of Integrative Medicine. 2018 Oct 1;23:84-9.

2.      Koolen HH, da Silva FM, Gozzo FC, de Souza AQ, de Souza AD. Antioxidant, antimicrobial activities and characterization of phenolic compounds from buriti (Mauritia flexuosa L. f.) by UPLC–ESI-MS/MS. Food Research International. 2013 May 1;51(2):467-73.

3.    Batista JS, Olinda RG, Medeiros VB, Rodrigues CM, Oliveira AF, Paiva ES, Freitas CI, Medeiros AD. Antibacterial and healing activities of buriti oil Mauritia flexuosa L. Ciência Rural. 2012;42(1):136-41.

4.      de OLIVEIRA AI, CABRAL JB, MAHMOUD TS, do NASCIMENTO GN, da SILVA JF, PIMENTA RS, de MORAIS PB. In vitro antimicrobial activity and fatty acid composition through gas chromatography-mass spectrometry (GC-MS) of ethanol extracts of Mauritia flexuosa (Buriti) fruits. Journal of Medicinal Plants Research. 2017 Oct 25;11(40):635-41.

 ·    If authors performed comparative study of antibacterial activity of different parts of the plants, the effectiveness of the study will surely be enhanced.

We understand that the comparative study is important, but since there are already some works demonstrating the activities of other parties, we centered our study in answering the antibacterial and modifying potential of the antibiotic activity of the fixed oil of the pulp that has stood out by the commercial value.

In the results and discussion we mention some works to enrich the data.

·   Table 3: GC/MS spectral analysis was performed to identify the phytochemicals constituents of the M. flexuosa fixed oil using the library NIST 08. Authors have just shown the table, but not the GCMS spectra. Authors must provide the GC/MS spectra that support the table 3. What are the other components present in pulp extract?

Table 3 reflects exactly what the GCMS chromatogram of the M. flexuosa fixed oil shows (attached). We are shipping the GCMS chromatogram as per request. The methodology used for GCMS analysis of the fixed oil allowed the identification of only two chemical constituents (palmitic acid: 10.19% and oleic acid: 89.81%).

Which is not surprising, depending on the analytical method used. In the work of Oliveira et al (Molecules 2011, 16, 5875-5885; doi: 10.3390 / molecules16075875) the analysis of the chemical composition of the M. flexuosa fixed oil by GCMS occurs identically, with values of 16.6% and 83.4% for palmitic and oleic acids, respectively.

In the work of Nobre et al. (European Journal of Integrative Medicine doi: 10.1016 / j .jim.2018.09.009) the same fact occurs, where the analysis of the chemical composition of the fixed oil of M. flexuosa by GC FID provided as majorities: palmitic acid (19.73%) and oleic acid (72.14%), besides some traces of: Lauric (0.05%); Myristic (0.08%); palmitoleic acid (0.4%).

·         Why authors have chosen Aminoglycoside antibiotics? Though Aminoglycoside antibiotics display bactericidal activity against Gram-negative aerobes and some anaerobic bacilli where resistance has not yet arisen but generally not against Gram-positive.

Dear reviewer, the choice to use amynoglicosides is due the high possibility of interaction between these drugs and natural products of several sources. This effect can be observed in our published articles (please, seek on pubmed using “Coutinho HD” and you could observe this effect. Due this fact, we choose this antibiotc class and in these papers, we can observe the synergistic effect over gram positive and gram negative bacteria.

·         The synergistic effects of oil with antibiotics are not clear. How many folds the efficacy of antibiotics increased in association with oils.  

The results were presented in table (table 05) and in the text was expressed the reduction percentage of MIC.

         Authors did not mention the mechanism and mode of action of M. flexuosa pulp oils.

The possible mechanism of oil was reported.

·         Scanning electron or transmission electron microscopic analysis can described the mechanism of pulp oils.

Dear reviewer, the TEM can demosntarte the effect of the oil on the bacterial cell, however, the aim o four manuscript is demonstarte the potential synergism between oil and antibiotc. I agree that this microscopy technique could be useful but with our results, we demonstrated this effect aimed by other protocols. By this fact, I hope your comprehension.

·         Authors only used MIC to assess the antibacterial test, though for plants extract, diffusion methods showing clear zone of inhibition is more acceptable.

Dear reviwer, as indicted by several papers and including by the CLSI, the disk diffusion method is an excellent method to screen several products or drugs to demonstarte na antibacterial potential. However, to a future possibility of therapeutical use, the micrdilution method is indicated due make possible determine also the MIC as the MBC of the studied drug or natural product.

Reviewer 2 Report

The authors show physicochemical characterization of a fixed pulp oil from Mauritia flexuosa as well their antibacterial synergic effect against S. aureus when is combined with Amikacin or gentamicin. Abstract and discussion must be improved.

General Comments

Abstract must be more specific according with results. The synergic effect of fixed Pulp oil combined was observed only in Staphylococcus aureus SA–ATCC 6538.

Discussion must be improved. There is not discussion about mechanism associated with synergic effect of fatty acids in S. aureus.

The results of MIC assays Figure 01 (Effect of fixed oil of Mauritia flexuosa (FOMf) on the activity of aminoglycoside antibiotics 177 against strains of Staphylococcus aureus – SA and Escherichia coli – EC.)  could be in a table.

Methodology (2.5.3.). Methodology reported is a modified CLSI protocol, it must be clarified. Reference 14 is wrong.

Minor changes

Ln 49:  “bactéria” must be corrected

Ln 73-75: The sentence “The pulp was obtained by manually removing the internal mesocarp from the fruit, and then 73 crushed in blender and lyophilized” is repeated twice.

Ln 106-107: “Shigella flexneri 106 EC–ATCC 12022” must be corrected

Ln 122: “1.0 uL/mL” must be corrected

Ln 157 and 169: “Table 03 and Table 04” must be corrected,  

Ln 69: “Exsicata” word must be revised

Ln 208 “effusio” word must be corrected

Author Response

REVIEWER 2

Comments and Suggestions for Authors

The authors show physicochemical characterization of a fixed pulp oil from Mauritia flexuosa as well their antibacterial synergic effect against S. aureus when is combined with Amikacin or gentamicin. Abstract and discussion must be improved.

General Comments

Abstract must be more specific according with results. The synergic effect of fixed Pulp oil combined was observed only in Staphylococcus aureus SA–ATCC 6538.

The result of the abstract was organized.

Discussion must be improved. There is not discussion about mechanism associated with synergic effect of fatty acids in S. aureus.

Discussion checked.

The results of MIC assays Figure 01 (Effect of fixed oil of Mauritia flexuosa (FOMf) on the activity of aminoglycoside antibiotics 177 against strains of Staphylococcus aureus – SA and Escherichia coli – EC.)  could be in a table.

Thanks for the suggestion. The results were presented in table (table 05).

Methodology (2.5.3.). Methodology reported is a modified CLSI protocol, it must be clarified. Reference 14 is wrong.

Dear reviewer, the reference was corrected. But the description is correct. I sorry if I can not understand this observation

Minor changes

Ln 49:  “bactéria” must be corrected.

Corrected.

Ln 73-75: The sentence “The pulp was obtained by manually removing the internal mesocarp from the fruit, and then 73 crushed in blender and lyophilized” is repeated twice.

Text removed.

Ln 106-107: “Shigella flexneri 106 EC–ATCC 12022” must be corrected

Corrected.

Ln 122: “1.0 uL/mL” must be corrected

Corrected.

Ln 157 and 169: “Table 03 and Table 04” must be corrected.

Corrected

Ln 69: “Exsicata” word must be revised

Corrected.

Ln 208 “effusio” word must be corrected

Corrected.

Round 2

Reviewer 1 Report

Authors must provide the GCMS histogram and methodolgy of GCMS

Author Response

REVIEWER 1

Authors must provide the GCMS histogram and methodolgy of GCMS

Dear reviewer, the metthodology of the gc-ms is described in the material and methdos section, in the sub heading 2.4.

About the chromatogram, it was inserted in the text as a figure.
